# Yield and Nitrogen Status of Maize (*Zea mays* L.) Fertilized with Solution of Urea—Ammonium Nitrate Enriched with P, Mg or S

Jadwiga Wierzbowska * , Stanisław Sienkiewicz  and Arkadiusz Światły

Department of Agricultural and Environmental Chemistry, University of Warmia and Mazury in Olsztyn, 10-719 Olsztyn, Poland
* Correspondence: jadwiga.wierzbowska@uwm.edu.pl

**Abstract:** UAN is a popular nitrogen fertilizer, broadly used in world agriculture. Research concerning the effects of this fertilizer is just as common. Contrarily, studies on the combined application of UAN with P, Mg or S are lacking. This fact has stimulated our study, undertaken in order to evaluate the effects of maize grown for grain and fertilized with UAN enriched with the macronutrients (P, Mg and S) on the crop's yields and nitrogen metabolism. The following nitrogen fertilizers were applied: UAN 32%N, UAN + S—26% N + 3% S, UAN + P (Medium)—26% N and 11% $P_2O_5$, UAN + P (Starter)—21% N and 18% $P_2O_5$, UAN + Mg—20% N + 4% Mg. Based on the results of chemical analyses and yields of maize, the following indicators of nitrogen use efficiency (NUE) were calculated: agricultural efficiency (AE), physiological efficiency (PE), internal N utilization efficiency (IE), reciprocal internal N utilization efficiency (RIE), grain share in N accumulation ($HI_N$), recovery of N from mineral fertilizers ($R_N$) and partial nitrogen balance (PNB). The highest grain yields were harvested after the application of UAN + S/UAN + Mg, and after the pre-sowing and top-dressing application of UAN or UAN + P (Medium). Values of all calculated nitrogen use efficiency indicators were more strongly dependent on the weather conditions, which determined volumes of maize yields in a given year, than on the applied fertilization.

**Keywords:** maize yield; UAN; P K; S; nitrogen management indicators





## 1. Introduction

Maize is one of the oldest plants domesticated by man, and nowadays one of the major crops in terms of cropped farmland. The global production of maize grain in 2019 reached 1148 million tons, being 3.3-fold higher than in 1970, which corresponded to an annual increase of 3.41% [1]. In Poland, the total area of farmland cropped with maize increased from 152,273 ha in 2000 (total production of 923,341 ton) to 664,950 ha in 2020 (production of 3,664,550 ton). Thus, the increase in the maize-cropped farmland and maize production of maize in Poland has been more rapid than elsewhere in the world [2].

Nitrogen (N) fertilization is one of the most important agrotechnical treatments that enables the farmer to obtain desired crop yields [3,4]. Treatments aiming at increasing yields of cereals, including maize, must focus on a more efficient use of nitrogen contained in mineral fertilizers. A yield increase per unit of applied N is particularly important because of the concern raised by excessive amounts of nitrogen forms in the environment [5,6]. According to Wang X. et al. [7], mutual interactions between soil, weather and crops have a considerable influence on yields in a region, and should be taken into consideration when optimizing nitrogen management.

Application of excessive doses of N fertilizers and improper fertilization methods both lead to less efficient use of this element by plants and its more intensive leaching [8]. Contemporary methods must take into account possible improvements in fertilizer application [9]. An example is the addition of urease inhibitors to UAN. Nikolajsen et al. [10] demonstrated that urease and nitrification inhibitors reduced the emission of $NH_3$, which resulted in a higher efficiency and increased uptake of N.

The availability of N is one of the chief determinants of maize productivity. Grain yield produced by maize depends on the intensity of photosynthesis, and N deficit typically has a negative effect on the efficiency of this process [11]. Adequate N management (source, doses and frequency of application) can enhance the yielding by maize. In their studies, Abbasi et al. [12] and Szulc et al. [13] demonstrated a significant effect on maize yields produced by the type of N fertilizer.

The initially slow growth of maize is sometimes due to temperatures being too low after the plant's emergence and the plant's inhibited uptake of water and nutrients, especially of phosphorus [14]. Moreover, the root system poorly developed at that time can supply the plant with only those nutrients that appear in adequately high concentrations in the soil substrate. An appropriately high P concentration in the soil is essential for a fast development of the maize root system, in addition to which it helps to mollify consequences of nutritional stress [15]. The dynamics of the early development of maize expressed by the accumulation of dry matter is limited most severely by the deficit of P in soil [16]. The concentration of phosphates is very low in comparison to other anions as the share of phosphate ions in the total amount of anions in the soil substrate is no more than a few percent [17].

Magnesium (Mg) in plant tissues appears in similar concentrations to those of phosphorus [18]. However, Mg is easily leached from acid soils and the competition of other cations means that the uptake of $Mg^{2+}$ by plant roots can be hindered. Unfortunately, farmers are not always aware of Mg deficiency, and therefore a shortage of this element is an increasingly more serious factor limiting plant production [19]. The availability of N in soil and its adequate uptake by plants are linked to high plant productivity, especially during the critical phases of plant growth and development [20]. An additional supply of N together with Mg can improve the productivity because Mg enhances the uptake of nitrogen.

Sulphur (S) plays an important role in the formation of chlorophyll and biosynthesis of proteins and lipids in plants [21]. Furthermore, good S supply has a positive influence on the uptake of other nutrients and efficiency of fertilization. The efficient use of NPK fertilizers and the economy of their application can suffer when S is deficient [22]. Sulphur is also beneficial for the plant growth parameters, yield structure elements and consequently for the yields of maize [23–25].

It seems that a good way to supply crops with available forms of P, Mg and S is by the enrichment of N fertilizers, especially UAN, with these elements. A hypothesis was put forward, suggesting that UAN with sulphur, magnesium or phosphorus can be used for fertilization of maize grown for grain, and the efficiency of such fertilization is comparable to the conventional application of fertilizers. Hence, this study was undertaken to test the effect of fertilization of maize with a solution of urea and ammonium nitrate (UAN) enriched with macronutrients (P, Mg and S) on maize yields and nitrogen management.

## 2. Materials and Methods

### 2.1. Description of the Experiment

A field experiment was conducted in 2015–2017. It was set up on production fields owned by the Production and Experimental Enterprise located in Bałcyny (51.6667° N, 18.1667° E). The surface area of a plot for harvest was 450 $m^2$. In each year, the experiment was set up on lessivé soil formed from medium clay [26]. It had the following parameters: slightly acid reaction, determined potentiometrically on a CP-505 pH meter (pH in 1 mol $dm^{-3}$ KCl ranged from 5.70 to 6.33), the C-org. content was between 12.5 and 1.32 g $kg^{-1}$ (Vario Max Cube CN Elementar apparatus), and the concentrations of available forms of P and K (by the Egner–Riehm method) and Mg (by the Schachtschabel method) were as follows: P—97.8–135.3; K—182.7–224.1; Mg—52.0–82.0, while the content of S-$SO_4^{2-}$ was within 4.0–14.0 mg $kg^{-1}$ [27]. The experiments were laid out in a random-block design with 4 replicates (Table 1). Maize was fertilized with nitrogen twice: before sowing (100 kg N $ha^{-1}$) and in the 14–16 BBCH stage (80 kg N $ha^{-1}$).

**Table 1.** Design of the experiment.

| Number of Object | Date of Application | |
| --- | --- | --- |
| | Pre-Sowing (100 kg N ha$^{-1}$) | 4–6 Leaf Phase (14–16 BBCH) (80 kg N ha$^{-1}$) |
| I. * | control, no nitrogen fertilization | |
| II. * | ammonium nitrate | urea |
| III. * | UAN | urea |
| IV. * | UAN | UAN |
| V.* | UAN + S | UAN + Mg |
| VI. ** | UAN + P (Medium) | UAN + P (Medium) |
| VII. *** | UAN + P (Starter) | UAN + S |
| VIII. *** | UAN + P (Starter) | UAN + Mg |

*—pre-sowing dose of phosphorus (37.36 kg P ha$^{-1}$) in the form of granulated fertilizer. **—pre-sowing dose of phosphorus (4.18 kg P ha$^{-1}$) in the form of granulated fertilizer. ***—pre-sowing fertilization with phosphorus in a liquid form.

The following nitrogen fertilizers were applied in the experiment: UAN—32%N, UAN+—26% N + 3% S, UAN + P (Medium)—26% N and 11% $P_2O_5$, UAN + P (Starter)—21% N and 18% $P_2O_5$, UAN + Mg—20% N + 4% Mg.

The maize cultivar sown in the experiment was the NK Borago (single hybrid), a variety with large, flex cobs, well-filled with glassy flint kernels. It is an early variety (FAO 220) with excellent vernal vigor, recommended for cultivation on good soils, which warm up in the spring less rapidly [28]. Plants of this maize cultivar are also distinguished by good health.

In 2015 and 2016, maize was seeded in the last ten days of April, while in 2017 it was sown on 1 May, in the amount of 90,000 germinating kernels per 1 hectare, in rows set 75 cm apart; the forecrop was winter wheat. In objects 2–8, nitrogen fertilization was applied in the quantity of 180 kg N ha$^{-1}$, split into two doses: 100 kg N ha$^{-1}$ before sowing (BBCH 00) and 80 kg N ha$^{-1}$ at the phase of 4–6 leaf (BBCH 14–16), in the forms defined in the design of the experiment. Before sowing, soil in objects 1–5 was enriched with 37.36 kg P ha$^{-1}$ (as triple superphosphate). Soil in object 6, before sowing maize, was enriched with 4.18 kg P ha$^{-1}$ in the solid form and 16.58 kg P ha$^{-1}$ as UAN-P Medium, while the remaining dose (16.58 kg P ha$^{-1}$) was applied as top-dressing fertilization mainly in the form of UAN-P Medium. In objects 7 and 8, the entire dose of phosphorus (37.36 kg P ha$^{-1}$) was applied in the liquid form (UAN+ P Starter). All the objects were supplied 160.02 kg K ha$^{-1}$ as 60% potassium salt before sowing the maize. At the 16–18 BBCH maize development phase, Insol Zn in a dose of 2 dm$^3$ ha$^{-1}$ (100 g Zn ha$^{-1}$) was applied to maize leaves. Weeds were controlled during the 3rd leaf unfolded stage (BBCH 13) by applying Lumax 537.5 SE in a dose of 3.5 dm$^3$ ha$^{-1}$. Maize harvest took place in the last ten days of October in the first and second year of the experiment, being delayed until the second ten days of November in the last year due to heavy rainfalls.

*2.2. Yield and N Content Determination*

10 maize plants were harvested from each plot, in order to determine the content of dry matter, content of N, and yield of straw. The plant material was divided into grain and straw (stems, leaves, rachis). The grain yield from each plot was determined by weight after threshing and then corrected to the standard moisture (14%). The yields of grain and straw were converted per 1 ha.

The harvest index (HI) was calculated from the formula:

$$HI = Y_G/(Y_G + Y_S) \tag{1}$$

where:

HI—harvest index
$Y_G$—grain yield (t ha$^{-1}$ d.m.)

$Y_S$—straw yield (t ha$^{-1}$ d.m.)

The N content in grain and straw was determined after mineralization of plant material in concentrated sulphuric acid ($H_2SO_4$) with hydrogen peroxide ($H_2O_2$) added as an oxidant (Speed Digester K-439; BÜCHI, Switzerland). The mineralized plant material was submitted to the Kjeldahl method (KjelFlex K-360; BÜCHI, Switzerland) to determine the N content [27].

*2.3. Methods of Calculating of Nitrogen Use Efficiency (NUE) Indicators*

Based on the results of chemical analyses and yields of maize, the fertilization efficiency indices were calculated [29]:

- agronomic efficiency (AE),

$$AE \text{ [kg kg}^{-1} \text{ N]} = (Y_N - Y_0)/D_N \tag{2}$$

where:

$Y_N$—yield of maize fertilized with N,
$Y_0$—yield of maize from the control object
$D_N$—N dose [kg ha$^{-1}$]

- physiological efficiency (PE),

$$PE \text{ (kg kg}^{-1} \text{ N)} = (YN - Y0)/(U_N - P_0) \tag{3}$$

where:

$Y_N$—yield of maize fertilized with N,
$Y_0$—yield of maize from the control object
$U_N$—nitrogen uptake by fertilized plants (kg N ha$^{-1}$),
$U_0$—nitrogen uptake by control plants (kg N ha$^{-1}$)

- internal N utilization efficiency (IE) [30]:

$$IE \text{ (kg kg}^{-1} \text{ N)} = Y/U \tag{4}$$

where:

IE—internal N utilization efficiency
Y—grain yield[t ha$^{-1}$]
U—nitrogen uptake by plants (kg N ha$^{-1}$)

- reciprocal internal N utilization efficiency (unit uptake):

$$RIE \text{ [kg 1000 kg}^{-1}\text{]} = (U/Y_G) \times 1000 \tag{5}$$

where:

RIE—reciprocal internal N utilization efficiency
U—nitrogen uptake by plants (kg ha$^{-1}$)
$Y_G$—grain yield (t ha$^{-1}$)

- share of grain in nitrogen accumulation (HI$_N$):

$$HI_N \text{ [%]} = (U_G/U) \times 100 \tag{6}$$

where:

HI$_N$—share of grain in nitrogen accumulation
$U_G$—accumulation of N in grain (kg N ha$^{-1}$)
U—accumulation of N in maize aerial parts (kg N ha$^{-1}$)

- 　　　N-fertilizer recovery efficiency ($R_N$) [29]:

$$R_N \ [\%] = [(U_N - U_0)/D_N] \times 100 \tag{7}$$

where:

$R_N$—recovery efficiency
$U_N$—nitrogen uptake by fertilized plants (kg N $ha^{-1}$)
$U_0$—nitrogen uptake by control plants (kg N $ha^{-1}$)
$D_N$—N dose (kg N $ha^{-1}$)

- 　　partial nitrogen balance (PNB):

$$PNB = U_G/D_N \tag{8}$$

where:

PNB—partial nitrogen balance
$U_G$—accumulation of N in grain (kg N $ha^{-1}$)
$D_N$—N dose (kg N $ha^{-1}$)

### 2.4. Statistical Analysis

The results underwent statistical processing by applying analysis of variance (ANOVA) in a STATISTICA 13® software package (1984-2017 TIBCO Software Inc.). The analysis of variance was performed as a 3-year series for one-way design. Differences between the means were compared with the Tukey's HSD post hoc test at significance $p < 0.05$. Correlation coefficients (r) were also calculated for the relationships between the selected parameters, analyzed in this experiment. In addition, cluster analysis including agglomerative clustering (single-linkage) was carried out, and the applied distance measure was the Euclidean distance.

### 2.5. Meteorological Conditions

In the first and third year of the experiment, the emergence of maize plants proceeded during a period of relatively modest rainfalls (40 and 55% of the multi-year average rainfalls, respectively). The best conditions for plant emergence were noted in the second year of the study. During the 2nd and 3rd research year, the weather conditions in June, that is the time when maize plants go through vegetative development, ensured adequate moisture and the air temperature was close to the long-term average. However, during the first year, the atmospheric precipitations were much below the long-term mean (Table 2).

**Table 2.** Characteristics of the meteorological conditions during the experiment.

| Month | Average Daily Air Temperature [°C] | | | Long-Term Average (1981–2010) | Rainfalls [mm] | | | Long-Term Average (1981–2010) | K *–Selyaninov Hydrothermal Coefficient [31] | | |
|---|---|---|---|---|---|---|---|---|---|---|---|
| | 2015 | 2016 | 2017 | | 2015 | 2016 | 2017 | | 2015 | 2016 | 2017 |
| April | 7.2 | 8.8 | 6.7 | 7.7 | 23.4 | 33.1 | 52.1 | 29.8 | 1.08 | 1.26 | 2.59 |
| May | 12.1 | 14.8 | 13.1 | 13.2 | 25.4 | 70.8 | 34.0 | 62.3 | 0.68 | 1.53 | 0.84 |
| June | 15.7 | 18.0 | 16.7 | 15.8 | 43.0 | 66.3 | 109.9 | 72.9 | 0.91 | 1.23 | 2.19 |
| July | 18.0 | 18.5 | 17.3 | 18.3 | 71.0 | 138.6 | 106.1 | 81.2 | 1.27 | 2.41 | 1.98 |
| Aug | 21.3 | 17.5 | 18.7 | 17.7 | 13.0 | 71.9 | 54.8 | 70.6 | 0.19 | 1.32 | 0.95 |
| Sept | 14.2 | 14.7 | 13.5 | 13.0 | 51.2 | 17.1 | 211.1 | 56.2 | 1.19 | 0.39 | 5.21 |
| Oct | 6.6 | 6.9 | 9.4 | 8.1 | 20.8 | 96.3 | 160.3 | 51.2 | 1.01 | 4.51 | 5.52 |

* K: 0–0.5—drought; 0.6–1.0—dry weather; 1.1–2.0—wet weather; >2.1—very wet weather.

In the first year of the experiment, when the lowest atmospheric precipitations during the vernal plant growth were noted, July was also characterized by the lowest value of the Selyaninov hydrothermal coefficient (K = 1.27) compared to the two subsequent years,

when it reached 2.41 and 1.98, respectively [31]. The end of maize flowering and beginning of the milk maturity stage occurred in August. During the first year, this month was marked by a drought, when the Selyaninov coefficient was barely 0.19, which was due to high air temperatures and low rainfall. During the second and third year of the experiment, in August, this coefficient reached 1.32 (humid) and 0.95 (dry weather).

In September, the mean daily temperature in 2015–2017 was approximately the same as the long-term one. In the first year of the study, the total atmospheric rainfall in September did not diverge from the average for 1981–2010, whereas the second year was dry (K = 0.39). September and October 2017 experienced the record high rainfalls, 211 mm and 160 mm respectively, which was 3-fold higher than the 1981–2010 average.

## 3. Results

Research on the effect of UAN on many agricultural crops is quite widespread. However, there are no studies on the combined application of UAN with P, Mg and S. This motivated us to launch an experiment with the aim of evaluating the impact of fertilization of maize grown for grain with UAN enriched with macronutrients (P, Mg and S), on the crop's yield and nitrogen metabolism.

### 3.1. Maize Yield and Harvest Index

The significantly lowest grain yield (5.26 t ha$^{-1}$) was harvested in the first year from the control object (Table 3). In the second year, similar grain yields (11.05–11.68 t ha$^{-1}$) were collected from all nitrogen-fertilized plots, regardless of the form of the fertilizer. Maize grain yield depended on both nitrogen fertilization and meteorological conditions during the year of maize cultivation. On average, the highest grain yield (11.22 t ha$^{-1}$) was harvested in the second year, being significantly lower in the first and third year (by 41.8 and 15%, respectively). Compared to the control, nitrogen fertilization significantly increased grain yield. The increase varied from 29% in the UAN + S/UAN + Mg object to 35% (maize fertilized before sowing and top-dressing with UAN or UAN + P (Medium)). Differences in the effect produced by the different fertilizers were not confirmed statistically as being significant.

**Table 3.** Yields of common maize (mean ± SE).

| Fertilization | Years of the Study | | | Average |
| --- | --- | --- | --- | --- |
| | **2015** | **2016** | **2017** | |
| | Grain yield (t ha$^{-1}$) | | | |
| Control, no N fertilization | 5.26 ± 0.31 a * | 8.61 ± 0.17 e | 7.38 ± 0.10 d | 7.08 ± 0.43 B |
| Ammonium nitrate/urea | 6.60 ± 0.09 b | 11.57 ± 0.14 h | 9.80 ± 0.09 fg | 9.32 ± 0.62 A |
| UAN/urea | 6.65 ± 0.09 b | 11.63 ± 0.09 h | 9.64 ± 0.09 f | 9.31 ± 0.62 A |
| UAN/UAN | 6.81 ± 0.12 bc | 11.72 ± 0.15 h | 10.21 ± 0.13 g | 9.56 ± 0.62 A |
| UAN + S/UAN + Mg | 6.42 ± 0.07 b | 11.05 ± 0.09 h | 9.56 ± 0.17 f | 9.16 ± 0.63 A |
| UAN + P (Medium)/UAN + P (Medium) | 6.77 ± 0.44 bc | 11.68 ± 0.11 h | 10.22 ± 0.19 g | 9.56 ± 0.64 A |
| UAN + P(Starter)/UAN + S | 7.23 ± 0.08 cd | 11.62 ± 0.14 h | 9.71 ± 0.12 f | 9.52 ± 0.55 A |
| UAN + P(Starter)/UAN + Mg | 6.57 ± 0.22 b | 11.40 ± 0.11 h | 9.75 ± 0.21 fg | 9.24 ± 0.61 A |
| Average | 6.54 ± 0.12 A | 11.22 ± 0.18 C | 9.54 ± 0.19 B | − |
| | Straw yield (t ha$^{-1}$) | | | |
| Control, no nitrogen fertilization | 7.33 ± 0.55 a–e | 12.95 ± 0.92 f | 6.73 ± 0.12 a–e | 9.00 ± 0.91 A |
| Ammonium nitrate/urea | 6.02 ± 0.62 a–c | 16.57 ± 2.03 gh | 8.34 ± 0.37 c–e | 10.31 ± 1.51 A |
| UAN/urea | 5.02 ± 0.36 a | 14.65 ± 0.72 fg | 8.70 ± 0.06 de | 9.46 ± 1.22 A |
| UAN/UAN | 8.04 ± 0.44 b–e | 15.04 ± 1.17 fg | 8.92 ± 0.35 e | 10.67 ± 1.02 A |
| UAN + S/UAN + Mg | 6.47 ± 0.83 a–d | 16.27 ± 1.51 gh | 7.20 ± 0.06 a–e | 9.98 ± 1.44 A |
| UAN + P (Medium)/UAN + P (Medium) | 5.77 ± 0.38 ab | 14.81 ± 0.91 fg | 7.84 ± 0.08 b–e | 9.47 ± 1.20 A |
| UAN + P(Starter)/UAN + S | 6.40 ± 0.61 a–d | 15.26 ± 1.30 f–h | 7.91 ± 0.16 b–e | 9.86 ± 1.25 A |
| UAN + P(Starter)/UAN + Mg | 6.36 ± 0.72 a–d | 17.47 ± 1.63 h | 8.40 ± 0.11 c–e | 10.74 ± 1.55 A |
| Average | 6.43 ± 0.24 A | 15.38 ± 0.48 C | 8.01 ± 0.14 B | − |

**Table 3.** *Cont.*

| Fertilization | Years of the Study | | | Average |
|---|---|---|---|---|
| | **2015** | **2016** | **2017** | |
| Harvest index (HI) | | | | |
| Control, no nitrogen fertilization | 0.38 ± 0.01 a | 0.40 ± 0.01 ab | 0.52 ± 0.01 g–j | 0.43 ± 0.02 B |
| Ammonium nitrate/urea | 0.48 ± 0.03 a–h | 0.41 ± 0.03 ab | 0.54 ± 0.01 ij | 0.48 ± 0.02 AB |
| UAN/urea | 0.53 ± 0.02 h–j | 0.44 ± 0.01 b–e | 0.52 ± 0.00 h–j | 0.50 ± 0.01 A |
| UAN/UAN | 0.42 ± 0.01 a–c | 0.44 ± 0.02 b–d | 0.53 ± 0.01 h–j | 0.46 ± 0.02 AB |
| UAN + S/UAN + Mg | 0.47 ± 0.03 c–f | 0.41 ± 0.02 ab | 0.57 ± 0.01 j | 0.48 ± 0.02 AB |
| UAN + P(Medium)/UAN+ P(Medium) | 0.50 ± 0.02 f–i | 0.44 0.01 b–e | 0.56 ± 0.01 j | 0.50 ± 0.02 A |
| UAN + P(Starter)/UAN + S | 0.48 ± 0.02 f–i | 0.43 ± 0.02 b–d | 0.55 ± 0.01 j | 0.49 ± 0.02 A |
| UAN + P(Starter)/UAN + Mg | 0.47 ± 0.02 d–g | 0.39 0.02 ab | 0.53 ± 0.00 h–j | 0.47 ± 0.02 AB |
| Average | 0.47 ± 0.01 B | 0.42 ± 0.01 A | 0.54 ± 0.00 C | − |

*—data marked with same letters do not differ significantly at $p < 0.05$.

The highest straw yield (17.47 t ha$^{-1}$) was produced by maize grown in the second year and fertilized with UAN + P (Starter) before sowing and with UAN + Mg (Table 3) by top-dressing; the lowest straw yield (5.02 t ha$^{-1}$) was harvested in the first year from the plot fertilized with UAN prior to sowing and with urea applied as a top-dressing fertilizer. In turn, the highest maize straw yield, on average 15.38 t ha$^{-1}$, was harvested in 2016. Due to dry spells of the weather in the first year, the average straw yield was 2.7-fold lower than in 2016.

The HI index informs about the share of grain in the aerial biomass produced by the plant. The significantly highest HI value (0.57) was calculated for the maize grown in the third year of the experiment and fertilized with UAN + S prior to sowing followed by UAN + Mg applied as a top-dressing fertilizer (Table 3). The lowest HI value (0.38) was noted in the first year and concerned the maize grown on the control plots. The HI values in the three years of the experiment were significantly varied. Regarding the years, the highest HI (0.54) was achieved in the third year, while the average HI in the first and second year was lower by 13 and 23%, respectively. N fertilization increased the share of grain in the aerial biomass of maize, but significantly higher HI values relative to the control were only obtained from the maize grown in the three treatments: UAN + P (Medium) applied prior to sowing and top-dressing, UAN prior to sowing and urea applied as a top-dressing fertilizer, and UAN + P (Starter) before sowing and UAN + S top-dressing application (HI = 0.5, 0.50 and 0.49, respectively).

*3.2. Nitrogen Content*

The significantly lowest N content (11.17 g kg$^{-1}$ d. m.) was determined in the grain of maize from the control object harvested in the third year of the trials (Table 4). The significantly highest N content (17.01 g kg$^{-1}$ d. m.) was accumulated in the grain of maize fertilized with solid N fertilizers (ammonium nitrate prior to sowing and urea as top-dressing) in the first year of the study. The N content in maize grain depended on both the form of fertilizers and meteorological conditions. The highest N concentration N (15.62 g kg$^{-1}$ d. m.) occurred in maize grain harvested in 2015, while the lowest content of this element in grain (20% less than in 2015) was accumulated in 2017. Compared to the control, the application of fertilizers led to an increase in the grain content of N from 7.85% (UAN before sowing and top-dressing) to 13.09% (maize fertilized with solid nitrogen fertilizers, i.e., pre-sowing application of ammonium nitrate and top-dressing application of urea).

**Table 4.** Nitrogen content (g kg$^{-1}$ d.m.) in maize grain and straw (mean $\pm$ SE).

| Fertilization | Years of the Study | | | Average |
|---|---|---|---|---|
| | 2015 | 2016 | 2017 | |
| Grain | | | | |
| Control, no N fertilization | 14.46 $\pm$ 0.06 d * | 14.47 $\pm$ 0.03 d | 11.17 $\pm$ 0.16 a | 13.37 $\pm$ 0.55 B |
| Ammonium nitrate/urea | 17.01 $\pm$ 0.01 m | 15.78 $\pm$ 0.05 k | 12.58 $\pm$ 0.07 bc | 15.12 $\pm$ 0.66 A |
| UAN/urea | 15.24 $\pm$ 0.11 gh | 15.55 $\pm$ 0.07 i–k | 12.48 $\pm$ 0.07 b | 14.42 $\pm$ 0.49 AB |
| UAN/UAN | 15.4 $\pm$ 0.09 2 h–j | 14.93 $\pm$ 0.11 ef | 12.69 $\pm$ 0.06 bc | 14.35 $\pm$ 0.42 AB |
| UAN + S/UAN + Mg | 15.31 $\pm$ 0.01 ghi | 14.83 $\pm$ 0.11 e | 12.82 $\pm$ 0.09 c | 14.32 $\pm$ 0.38 AB |
| UAN + P(Medium)/ UAN + P(Medium) | 16.74 $\pm$ 0.01 l | 15.13 $\pm$ 0.02 fg | 12.61 $\pm$ 0.02 bc | 14.83 $\pm$ 0.60 A |
| UAN + P(Starter)/UAN + S | 15.24 $\pm$ 0.22 gh | 15.47 $\pm$ 0.11 h–j | 12.77 $\pm$ 0.05 c | 14.49 $\pm$ 0.44 AB |
| UAN + P(Starter)/UAN + Mg | 15.57 $\pm$ 0.12 jk | 15.4 $\pm$ 0.11 2 h–j | 12.83 $\pm$ 0.05 c | 14.61 $\pm$ 0.45 AB |
| Average | 15.62 $\pm$ 0.17 C | 15.20 $\pm$ 0.09 B | 12.49 $\pm$ 0.11 A | – |
| Straw | | | | |
| Control, no N fertilization | 8.00 $\pm$ 0.08 b | 10.21 $\pm$ 0.07 g | 7.09 $\pm$ 0.01 a | 8.44 $\pm$ 0.46 B |
| Ammonium nitrate/urea | 11.09 $\pm$ 0.02 j | 12.55 $\pm$ 0.03 l–n | 10.67 $\pm$ 0.07 i | 11.44 $\pm$ 0.29 A |
| UAN/urea | 9.85 $\pm$ 0.10 d–f | 12.10 $\pm$ 0.01 k | 9.71 $\pm$ 0.01 d | 10.55 $\pm$ 0.39 A |
| UAN/UAN | 9.96 $\pm$ 0.04 f | 12.37 $\pm$ 0.04 l | 9.77 $\pm$ 0.04 de | 10.70 $\pm$ 0.42 A |
| UAN + S/UAN + Mg | 9.33 $\pm$ 0.03 c | 12.69 $\pm$ 0.14 n | 9.46 $\pm$ 0.03 c | 10.49 $\pm$ 0.55 A |
| UAN + P(Medium)/UAN + P(Medium) | 9.84 $\pm$ 0.09 def | 12.49 $\pm$ 0.06 lm | 9.78 $\pm$ 0.00 de | 10.70 $\pm$ 0.45 A |
| UAN + P(Starter)/UAN + S | 10.37 $\pm$ 0.05 gh | 12.67 $\pm$ 0.05 n | 9.77 $\pm$ 0.03 de | 10.94 $\pm$ 0.44 A |
| UAN + P(Starter)/UAN + Mg | 10.41 $\pm$ 0.05 h | 12.64 $\pm$ 0.12 mn | 9.89 $\pm$ 0.02 ef | 10.98 $\pm$ 0.42 A |
| Average | 9.86 $\pm$ 0.18 A | 12.22 $\pm$ 0.16 B | 9.52 $\pm$ 0.20 A | – |

* data marked with same letters do not differ significantly at $p < 0.05$.

The significantly least N (7.09 g kg$^{-1}$ d.m.) was in the straw of maize harvested from the control object in 2017, whereas in 2016 straw from the objects UAN + S/UAN + Mg, UAN + P (Starter)/UAN + S and UAN + P (Starter)/UAN + Mg contained over 78% more of this element (Table 4). The significantly most N (12.22 g kg$^{-1}$ d.m.) was determined in the straw of maize harvested in the second year of the experiment, while the average straw content of N in the other two years was lower by 19.31 and 22.09%, respectively. Same as in grain, the least N was found in the straw of maize not fertilized with this element (8.44 g kg$^{-1}$ d.m.). N fertilization significantly increased the content of N in straw (from 24.29%—UAN + S before sowing and top-dressing with UAN + Mg, to 35.54%—maize fertilized with ammonium nitrate before sowing and urea as a top-dressing application). No significant differences were confirmed between the N content in straw of maize fertilized with different forms of nitrogen-containing fertilizers.

### 3.3. Nitrogen Use Efficiency (NUE) Indicators

AE in the three years of our experiment varied from 8.15 to 16.55 kg kg$^{-1}$ N (Figure 1a). The highest AE was determined in the second year of the experiment, when the meteorological conditions were favorable for obtaining the highest maize grain yield. In turn, the lowest AE appeared in the first year, when the smallest grain yields were harvested due to a considerable rainfall shortage.

The data illustrated in Figure 1b show that the highest AE (13.89 kg kg$^{-1}$ N) was achieved in the plot fertilized with UAN before sowing and by top-dressing, and in the ones treated with UAN + P (Medium) prior to sowing and by top-dressing. The lowest net AE (11.56 kg k$^{-1}$ N) was determined for maize plants fertilized before sowing with UAN + S, and by top-dressing with UAN + Mg.

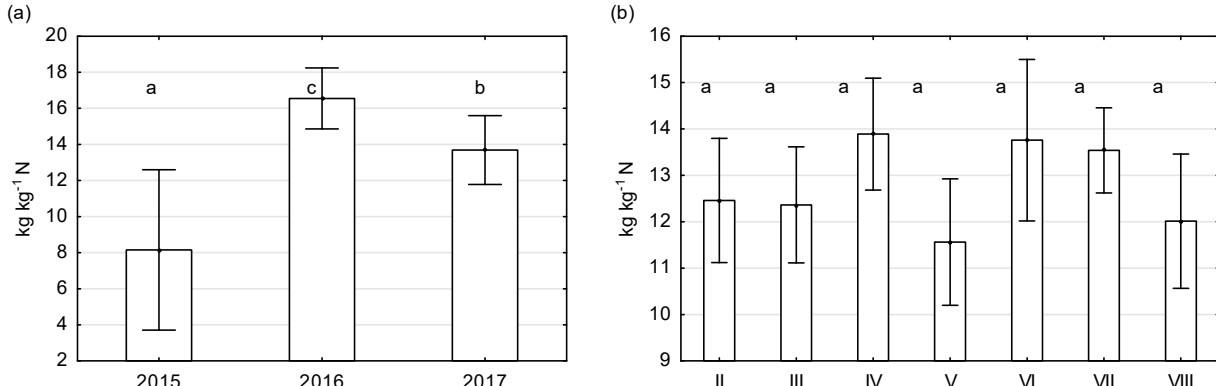

**Figure 1.** Agronomic efficiency (AE) of fertilizing maize depending on a study year (**a**) and applied fertilization (**b**)—mean ± SE; data marked with same letters do not differ significantly at $p < 0.05$; figure (**a**) years of experiment (2015, 2016 and 2017), figure (**b**) number of object (II–VIII)—see Table 1.

The highest PE (59.04 kg kg$^{-1}$ N) was determined in the first year of the experiment (Figure 2a), which had less rainfall than the 1981–2020 average, and the grain yields were smaller than the ones obtained in the entire experimental period. The lowest PE (27.19 kg kg$^{-1}$ N) was obtained in the second year of the study, which was characterized by higher rainfall than the long-term average, and the grain yield harvested was the highest in the whole experiment.

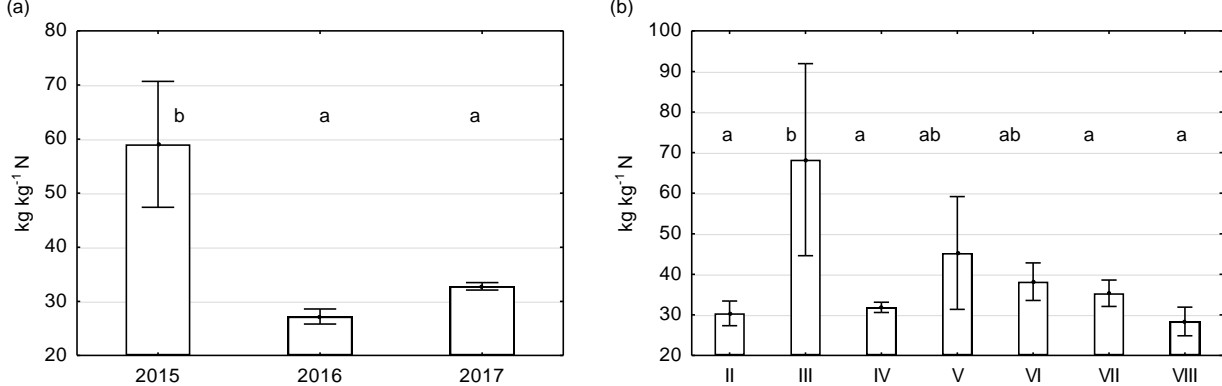

**Figure 2.** Physiological efficiency (PE) depending on a study year (**a**) and applied fertilization (**b**)—mean ± SE; data marked with same letters do not differ significantly at $p < 0.05$; figure (**a**) years of experiment (2015, 2016 and 2017), figure (**b**) number of object (II–VIII)—see Table 1.

Depending on the tested fertilization, PE varied from 28.38 to 68.27 kg kg$^{-1}$ N (Figure 2b). It was the lowest in the object fertilized prior to sowing with UAN + P(Starter) and top-dressed with UAN + Mg. The highest value of this parameter was achieved in the maize grown on the plots fertilized with UAN before sowing and with urea in a top-dressing application.

In this experiment, the IE values ranged from 31.5 to 49.0 kg kg$^{-1}$ N (Figure 3a). The lowest IE was obtained in 2016, when the meteorological conditions were optimal for the development and yielding of maize. In turn, the highest IE was achieved in the excessively wet 2017. On average for the three years of the study, the highest IE value was obtained from the control treatment (43.1 kg kg$^{-1}$ N), while the lowest one (37.7 kg kg$^{-1}$ N) originated from the treatment where maize was fertilized with solid N fertilizers (Figure 3b). The IE of maize fertilized with UAN or UAN with added macronutrients varied from 38.8 to 40.6 kg kg$^{-1}$ N. Variable weather conditions, and hence different maize yields in the particular research years, rendered the observed differences statistically insignificant.

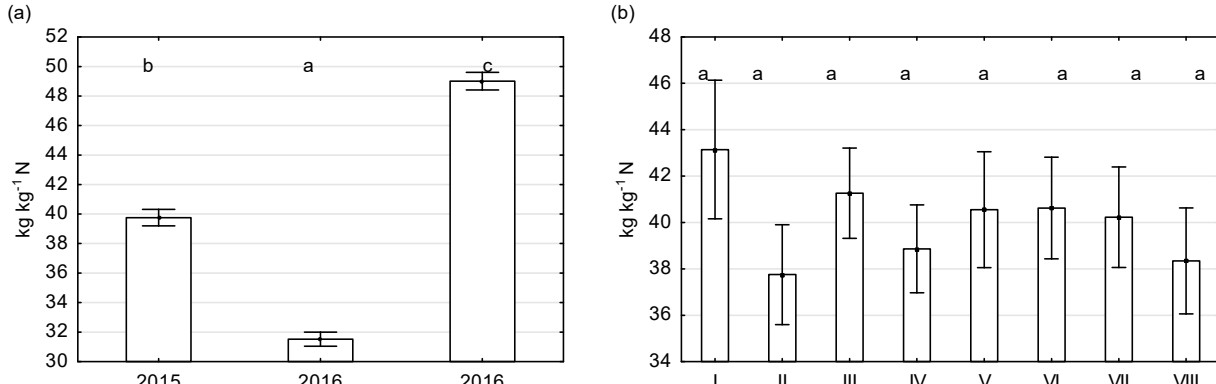

**Figure 3.** Internal nitrogen utilization efficiency (IE) depending on a study year (**a**) and applied fertilization (**b**)—mean ± SE; data marked with same letters do not differ significantly at $p < 0.05$; figure (**a**) years of experiment (2015, 2016 and 2017), figure (**b**) number of object (I–VIII)—see Table 1.

The IRE values were statistically differentiated in the research years (Figure 4a). The highest RIE (31.97 kg 1000 kg$^{-1}$) occurred in the second year of the experiment, which was characterized by the highest maize straw and grain yields. The lowest RIE (20.49 kg 1000 kg$^{-1}$) was determined in the third year, which was excessively wet. According to the average results for the entire study period, the tested fertilizers did not have a significant effect on the uptake of nitrogen per 1 ton of grain with the adequate quantity of straw, which varied from 24.34 to 27.53 kg 1000 kg$^{-1}$ (Figure 4b). The smallest RIE was determined in the control (no nitrogen fertilization), while the highest one occurred in the treatment where ammonium nitrate was applied before sowing and urea as a top-dressing fertilizer.

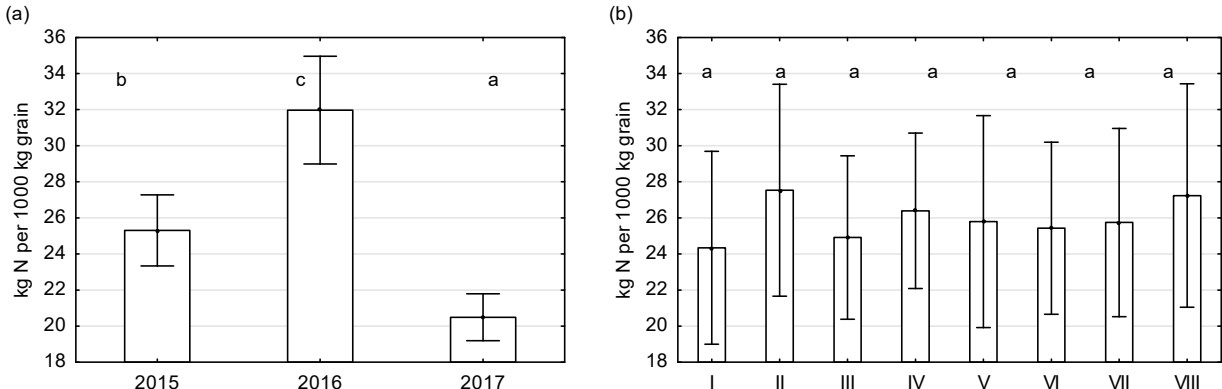

**Figure 4.** N uptake by the yield of 1000 kg of grain (RIE) including the appropriate amount of straw depending on the year of research (**a**) and fertilization (**b**)—mean ± SE; figure (**a**) years of experiment (2015, 2016 and 2017), figure (**b**) number of object (I–VIII)—see Table 1.

The average share of grain in N accumulation (HI$_N$) in the second year of the experiment was significantly lower (around 47%) than in the first and third years of the study (62 and 61%, respectively) (Figure 5a). The tested fertilization had no significant effect on the value of the HI$_N$ (Figure 5b). The HI$_N$ varied from approximately 55% (maize fertilized before sowing and then top-dressing with UAN, and maize treated with UAN + P (Starter) prior to sowing and with UAN + Mg applied as a top-dressing treatment) to about 60% after pre-sowing and top-dressing applications of UAN + P (Medium).

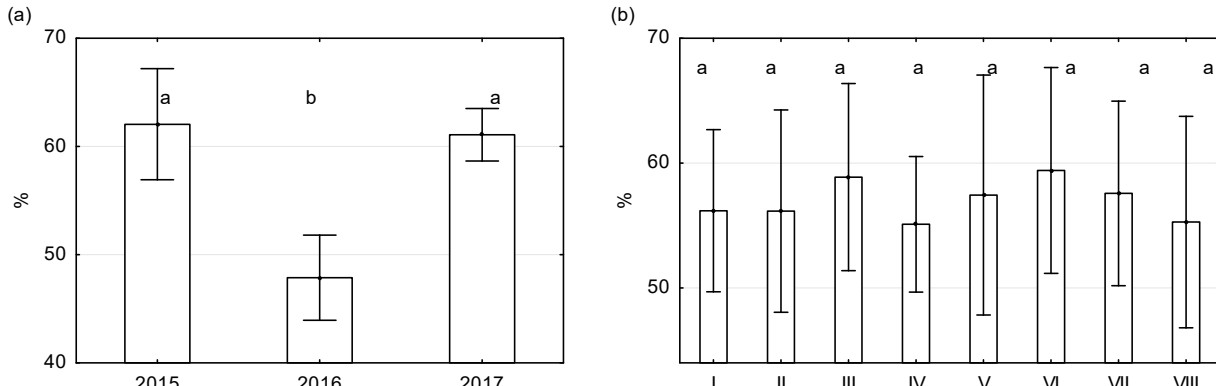

**Figure 5.** Share of maize grain in N accumulation (HI$_N$) depending on a study year (**a**) and applied fertilization (**b**)—mean ± SE; data marked with same letters do not differ significantly at $p < 0.05$; figure (**a**) years of experiment (2015, 2016 and 2017), figure (**b**) number of object (I–VIII)—see Table 1.

In this study, the recovery of N from fertilizers (R$_N$) ranged from 9 to 77% (Figure 6) The lowest R$_N$ was observed in the first year in the treatment fertilized with UAN before sowing and with urea as a top-dressing application. The highest R$_N$ was determined in the second year of the experiment, in the treatment fertilized with UAN + P (Starter) before sowing and with UAN + Mg top-dressing.

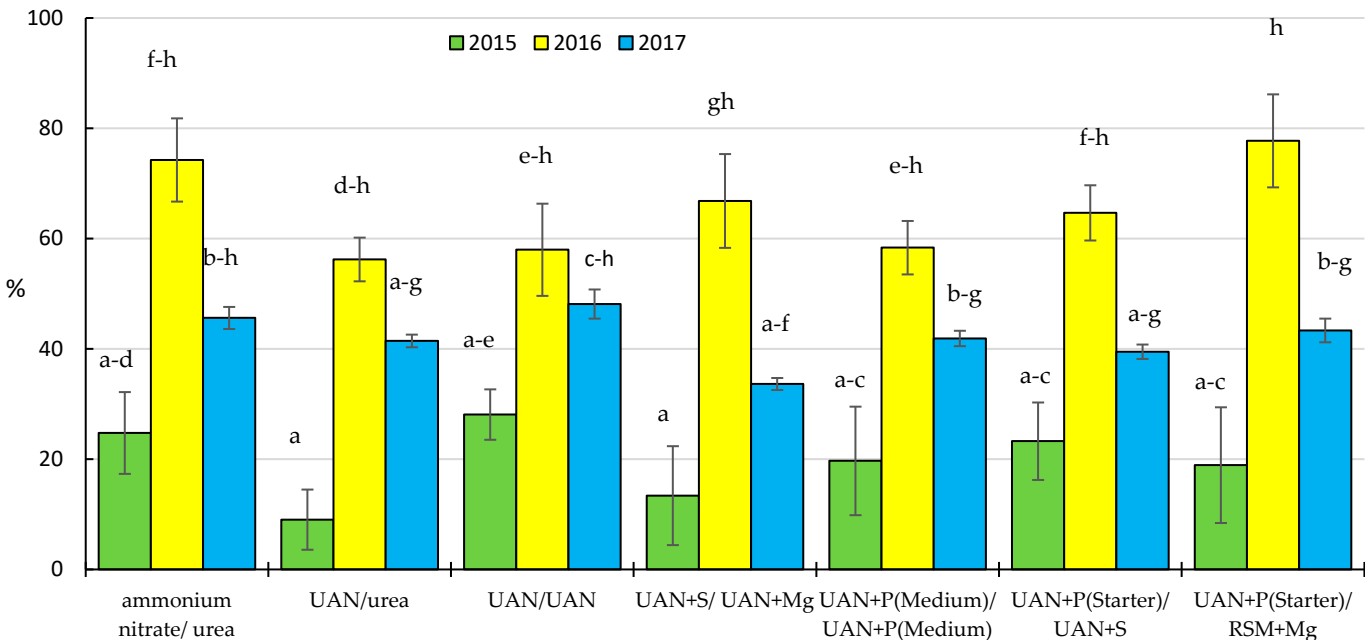

**Figure 6.** Recovery of N from mineral fertilizers (R$_N$) in the years of the experiment depending on the applied fertilization (mean ± SE; data marked with same letters do not differ significantly at $p < 0.05$), years of experiment (2015, 2016 and 2017) and treatments (see Table 1).

The R$_N$ in the years of the study was significantly varied (Figure 7a). The highest R$_N$ (62%) was recorded in the second year of the trials. In turn, the lowest R$_N$ (20%) appeared in the first year, in which the maize yield was the lowest because of a considerable shortage of rainfall. On average for the entire experiment, the lowest R$_N$ (35%) was noted in the object fertilized with UAN before sowing and with urea by top-dressing (Figure 7b). The highest value (48%) of this indicator was achieved when fertilizing maize with UAN + P (Starter) before sowing and with UAN + Mg by top-dressing. However, because of high fluctuations between the research years, differences in the R$_N$ between the treatments were not significant statistically.

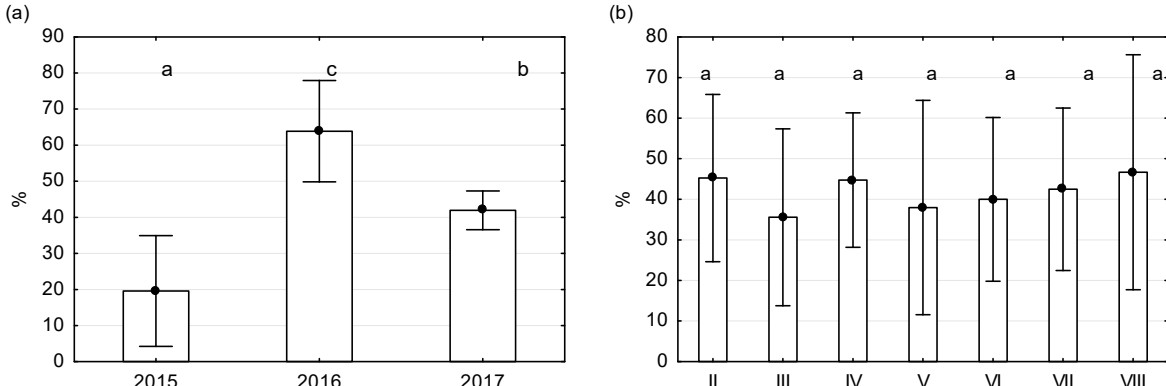

**Figure 7.** Recovery of N from fertilizers (R$_N$) depending on the years of the study (**a**) and applied fertilization (**b**)—mean ± SE; data marked with same letters do not differ significantly at $p < 0.05$; figure (**a**) years of experiment (2015, 2016 and 2017), figure (**b**) number of object (II–VIII)—see Table 1.

The weather conditions that varied between the years meant that the PNB values were different in the subsequent years (Figure 8a). In the dry year 2015 and in the very wet 2017, the low values of the PNB were observed (0.59 in 2015 and 0.69 kg kg$^{-1}$ in 2017). However, in the conditions that were optimal for the development of maize, such as in 2016, the BNP value reached 0.99 kg kg$^{-1}$. On average for the three years of the experiment (Figure 8b), depending on the applied fertilization regimes, PNB ranged from 0.72 (fertilization with UAN + S/UAN + Mg) to 0.78 kg kg$^{-1}$ (UAN + P (Medium)/UAN + P (Medium).

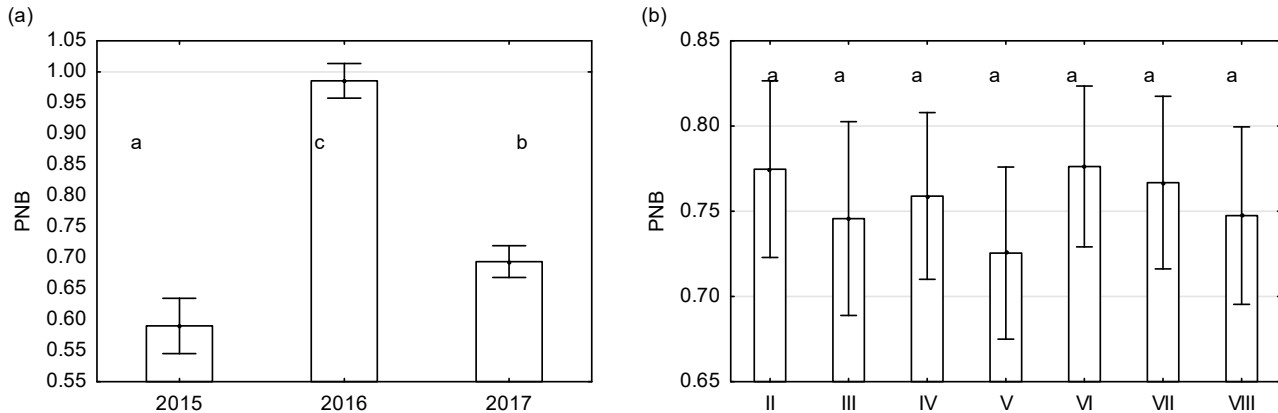

**Figure 8.** Partial nitrogen balance (PNB) depending on the years of the experiment (**a**) and applied fertilization (**b**)—(mean ± SE; data marked with same letters do not differ significantly at $p < 0.05$; figure (**a**) years of experiment (2015, 2016 and 2017), figure (**b**) number of object (II–VIII)—see Table 1.

Based on the results, significant relationships of NUE indicators with maize grain yields were determined (Table 5). The PNB was most strongly correlated with maize grain yield ($r = 0.91$ **). The correlations with grain yields determined for R$_N$ and AE were only slightly less strong ($r = 0.84$ ** and $r = 0.83$ **, respectively). A positive correlation was also detected for RIE ($r = 0.42$ **). On the other hand, the share of grain in nitrogen accumulation (HI$_N$) and PE were significantly negatively correlated with maize grain yield ($r = -0.72$ ** and $r = -0.35$ **, respectively).

**Table 5.** Matrix of correlation indices for yield and efficiency indices, n = 84.

| Trait | HI | HI$_N$ | RIE | IE | PE | AE | R$_N$ | PNB |
|-------|------|---------|---------|---------|---------|---------|---------|---------|
| Y$_G$ | −0.25 * | −0.72 ** | 0.42 ** | −0.34 ** | −0.35 ** | 0.83 ** | 0.84 ** | 0.91 ** |
| HI | 1.00 | 0.83 ** | −0.93 ** | 0.94 ** | 0.12 | −0.20 | −0.52 ** | −0.52 ** |
| HI$_N$ | | 1.00 | −0.87 ** | 0.82 ** | 0.30 * | −0.57 ** | −0.84 ** | −0.83 ** |
| RIE | | | 1.00 | −0.98 ** | −0.19 | 0.31 ** | 0.60 ** | 0.72 ** |
| IE | | | | 1.00 | 0.15 | −0.26 * | −0.53 ** | −0.67 ** |
| PE | | | | | 1.00 | −0.20 | −0.33 ** | −0.31 ** |
| AE | | | | | | 1.00 | 0.85 ** | 0.75 ** |
| R$_N$ | | | | | | | 1.00 | 0.83 ** |

**.  * indicate significant differences at $p < 0.01$. and $p < 0.05$. respectively. Key: Y$_G$—yield grain; HI—harvest index; HI$_N$—share of grain in nitrogen accumulation; RIE—N uptake by the yield of 1000 kg of grain; IE—internal nitrogen utilization efficiency; PE—physiological efficiency; AE—agronomic efficiency; R$_N$—recovery of N from fertilizers; PNB—partial nitrogen balance.

The cluster analysis enabled us to identify groups of fertilizers according to the effect they had on yield and on nitrogen management by common maize (Figure 9). Yielding and nitrogen management by maize fertilized with the tested fertilizers were distinctly different from the ones by maize grown in the control treatment (without N fertilization). The fertilization regimes that were the closest in terms of effects were the variants of pre-sowing UAN + S and top-dressing UAN + Mg and both pre-sowing and top-dressing with UAN + P (Medium). The second cluster was composed of the treatments fertilized with solid mineral fertilizers (ammonium nitrate/urea) or treated pre-sowing and top-dressing with UAN, but the effects produced by these fertilization variants were more diverse.

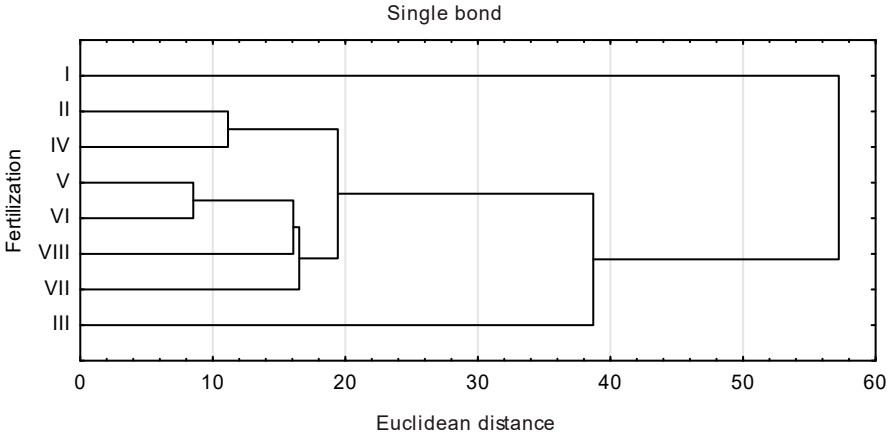

**Figure 9.** A dendrogram illustrating the effect of fertilization on yield and nitrogen management by maize (I–VIII—number of object, see Table 1).

## 4. Discussion

### 4.1. Maize Yield and Harvest Index

In this experiment, the high variation in maize yields was due to the variable weather conditions. Many authors draw attention to the fact that rainfalls and temperatures significantly affect the grain yield of maize [32–34]. Thus, the implementation of different nitrogen management strategies in maize production can be difficult because of the unpredictable course of the weather in each year [35]. According to Cofas [36], productivity of maize differs depending on the year when it is grown, and this is significantly affected by the variability of the climatic conditions, particularly in view of projected extreme climate events. Tremblay et al. [37] maintain that the yield-forming effect of N fertilization of maize can be enhanced by abundant and properly time-distributed rainfall as well as an adequate amount of warmth.

It can therefore be concluded that although we ensured that maize received sufficient quantities of nutrients in our experiment, its yield-forming potential was strongly limited by the dry weather in 2015. This conclusion can be supported by the results reported by



Srinivasan et al. [38], who claim that maize is particularly sensitive to drought during the flowering stage. Furthermore, Szulc and Bocianowski [39] determined that the type of N fertilizer and a dose of Mg do not differentiate significantly the dynamics of maize's early growth and development, manifested by dry matter accumulation. Yield of maize and use of N by this crop were comparable after the application of polymer-coated urea or urea and ammonium nitrate solution, but in years with dry spells the accumulation of N in the plant was much higher after it had been fertilized with UAN [40]. Grzebisz et al. [41] inform that maize fertilized with Zn was able to increase the N uptake in two different development stages (BBCH 17–19 and BBCH 75–87). The effect of Zn in the former stage was manifested by an increase in the accumulation of N, which prolonged the stage of intensive growth of tissues and/or organs. In the reproductive development stage of maize, the plants well-nourished with Zn accumulated N more rapidly, which was the basic condition for increased dry matter accumulation. In their study, Szulc et al. [42] found a significant effect on the maize grain yield increase produced by the application of Mg and S; namely, grain yield was higher by 5.7% to 10.7%. Fertilization of maize with N and S fertilizers (N:S ratio equal 5:1) had a beneficial effect on the morphological traits and yield structure elements as well as on the production of dry matter, which eventually had some influence on the yield of grain and straw [43]. A study carried out by Tabak et al. [44] confirms the positive effect of nitrogen and sulphur fertilization on yield of winter wheat.

Same as the grain yield, the yield of maize straw was determined by the weather conditions during the plant growing season. In the year 2016, which was favorable for the growth of the crop, the straw yield was nearly 2.5-fold higher than in the dry and hot year 2105, and over 90% higher than in 2017. Likewise, very high maize straw yields, between 10.5 and 12.1 t ha$^{-1}$, were obtained by Mickiewicz and Wróbel [45], who grew the crop in a monoculture in different soil tillage systems and fertilized it with boron and zinc by top-dressing. Maize fertilized with fresh litter from broiler chickens produced 9.67 t ha$^{-1}$ of straw, which was 3-fold more than maize grown without fertilization [46].

The HI of maize fertilized with increasing N doses in the experiment conducted by Kruczek [47] ranged from 0.52 (the object fertilized with 180 kg N·ha$^{-1}$) to 0.59 (the control object, with no fertilization).

### 4.2. Nitrogen Content

The N content in maize grain and straw determined in our study was similar to the amounts of this element given in references (Table 4). In a study by Filipek-Mazur et al. [48], the N content of maize grain was from 10.2 to 13.9 g kg$^{-1}$. Similar concentrations (11.0–13.1 g kg$^{-1}$) of N in maize grain were reported by Barczak et al. [49]. These researchers did not confirm a significant effect of the type of soil on the nitrogen content in maize grain. Much higher N concentrations (16.82–17.88 g kg$^{-1}$) in maize grain were demonstrated by Szulc et al. [50] and Baran et al. [51]. Depending on the dose and method of the application of P fertilizers, the N content of maize grain varied from 16.82 to 17.27 g kg$^{-1}$ [50]. In general, more N accumulated in the grain of maize from objects fertilized with ammonium phosphate than from ones receiving superphosphate. The content of protein in maize grain depended on a N dose more than on the type of N fertilizer [52]. Fertilization with S caused a small increase in the N content in maize grain [48]. However, S fertilization of maize grown on luvisol, podzol and black earth soils led to an increase in the grain content of N, unlike in maize grown on chernozem, where it resulted in a decrease of this element in grain [49]. Literature data prove that the N content in maize straw can vary within a wide range [50,51]. According to Baran et al. [51], the average N content in maize straw was 8.18 g kg$^{-1}$. In a study carried out by Szulc et al. [50], the N content of maize straw ranged from 0.19 to 10.46 g kg$^{-1}$. Moderate N fertilization (40 kg ha$^{-1}$) was conducive to the accumulation of N in straw, while large doses of this element (100 kg ha$^{-1}$) led to a decrease in the N content in maize straw.

*4.3. Nitrogen Use Efficiency Indices*

Agronomic efficiency (AE) reflects the direct effect of N fertilization on maize grain yield. In this study, the AE indicator for maize fertilized with 180 kg N ha$^{-1}$ ranged from 8.15 to 16.55 kg grain per 1 kg of applied nitrogen and depended on the grain yield, which varied in the three experimental years. Gołębiewska and Wróbel [53], who analyzed the effect of N dose on the efficiency of nitrogen fertilization in maize cultivation, achieved a grain yield increase from 9.59 kg (after the application of 270 kg N ha$^{-1}$) to 29.00 kg (after the application of 30 kg N ha$^{-1}$) per 1 kg of N supplied with the fertilizers. In another study, conducted by Kruczek [47], depending on the dose of N, the AE ranged from 31 kg grain (dose 45 kg N ha$^{-1}$) to 12.5 kg kg$^{-1}$ N (dose 180 kg N ha$^{-1}$) per 1 kg of applied N. The N application method also affected the AE of fertilization. After the foliar application of 45 kg N ha$^{-1}$, the AE was 18.8 kg kg$^{-1}$ N, but when the same amount of N was applied to soil, the AE value reached 21.3 kg kg$^{-1}$ N. Szmigiel et al. [54] studied the effect of organic and mineral fertilization on maize yields and obtained AE equal 43.3 kg (fertilization with 6 t ha$^{-1}$ of vermicompost) down to 17.5 kg kg$^{-1}$ N (fertilization with 120 kg N ha$^{-1}$ in the form of mineral fertilizers).

The PE of nitrogen fertilization informs about the increase in grain yield per 1 kg of N taken up by the plants. The reference values of the PE for cereals, corresponding to a good N fertilization balance, are within 30 to 60 kg kg$^{-1}$ [55]. Lower PE values implicate a reduction in the yielding due to the deficiency of nutrients, water stress or thermal stress, or toxicity of some agricultural chemicals applied [29]. In our experiment, PE was negatively correlated (r= $-0.35$**, $p < 0.05$) with grain yield (Table 5). In 2015, distinguished by a very large rainfall deficit, admittedly the lowest yields were harvested, but the N accumulation in grain was also relatively low, although this translated into the highest PE value (59.04 kg kg$^{-1}$ N) in the entire cycle of the trials. On the other hand, the highest grain yield harvested in the year 2016, which was meteorologically the most beneficial for maize growth, was not reflected in the value of the PE indicator (27.19 kg kg$^{-1}$ N), because maize accumulated over 50% of N in straw (for comparison, this percentage was merely 40% in 2015). On average for the entire experiment, the fertilization with UAN/urea had a particularly beneficial effect on the PE (68.27 kg kg$^{-1}$ N). According to Carneiro al. [56], PE is an indicator that varies over a wide range depending on the weather conditions during the growth and development of maize. This dependence was fully confirmed in our study (Figure 2a). In another experiment, completed by Carneiro al. [56], PE ranged from 39.6 to 29.9 kg kg$^{-1}$. Kruczek [47] determined the PE values between 61.7 (fertilization with 45 kg N ha$^{-1}$) and 59.3 kg kg$^{-1}$ N (fertilization with 135 kg N ha$^{-1}$). After an application of 45 kg N ha$^{-1}$ to soil cropped with maize, PE reached 62.4, whereas the top-dressing of maize plants with the same dose of N led to PE equal 58.8 kg kg$^{-1}$. The PE of maize fertilized with fermented bovine manure was as high as 105.5 kg grain per 1 kg of N taken up by the plants [46]. Tabak et al. [44] report on the beneficial effect of fertilizers containing N and S in proportions suitable for cereal production on the AE and PE in cultivation of winter wheat.

Internal N utilization efficiency (IE) is defined as a ratio of main yield to total uptake of this element, and demonstrates the plant's capacity to convert N obtained from different sources into commercial yield, e.g., grain [57]. The value of IE depends on the genotype of a crop, the environment in which it is grown and the mode of fertilization. A very high IE informs about the deficiency of a given nutrient. Low values of IE suggest poor internal conversion of nutrients due to stresses caused by deficits of other nutrients, drought stress, thermal stress or invasion of phytopathogens. Values of IE between 30–90 kg kg$^{-1}$ N are commonly noted for cereals, while values within 55–65 kg kg$^{-1}$ N are thought to be optimal [29]. In our study, the IE values did not exceed 50 kg kg$^{-1}$ and the indicator was negatively correlated with the grain yield of maize (r = $-0.34$ **).

Reciprocal internal N utilization efficiency (RIE) is defined as the amount of a nutrient in a plant needed to produce 1000 kg grain. According to Wrońska et al. [58], and same as in this study, the factor that most strongly differentiated RIE was the course of the

weather in particular years of the study (from 29.92 to 35.39 kg 1000 kg$^{-1}$). Depending on a N dose, IRE values ranged from 32.27 (fertilization with 80 kg N ha$^{-1}$) to 33.04 kg 1000 kg$^{-1}$ (fertilization with 160 kg N ha$^{-1}$). The cited authors maintained that the date of zinc application (BBCH0 and BBCH3) did not have a considerable effect on the amount of nitrogen taken up by the plant to produce 1000 kg of grain (32.80 and 32.51 kg N, respectively). However, compared to the control, the zinc fertilization depressed the uptake of nitrogen per unit (33.59 and 31.03 kg N 1000 kg$^{-1}$ of grain, respectively). In a study where maize was fertilized with different types of manure, Wieremiej [46] found that the value of IRE was from 16.19 kg t$^{-1}$ (fertilization with fermented bovine manure) to 25.56 kg t$^{-1}$ (fertilization with egg-laying hen manure). Jiang et al. [59] reported that the demand for nutrients increased until the target yield reached approximately 60–70% of the potential yield. Maize needed 16.7 kg N to produce 1000 kg of grain and the IE value for nitrogen was 60.0 kg·kg$^{-1}$ N.

The HI$_N$ indicator is shaped by the amount of N accumulated in maize grain and straw. The removal of N increased proportionally to the dose of this nutrient applied in maize fertilization [60]. In his study, Potarzycki [61] determined that HI$_N$ ranged from 71 to 77%. A wider range of this indicator was obtained by Wieremiej [46], concluding that HI$_N$ varied from 52 to 72%. In turn, Baran et al. [51] showed that grain accumulated around 65% of the N taken up by maize plants.

In this study, the value of the R$_N$ indicator ranged from 9 to 77% and depended mostly on the course of the weather conditions and the volume of harvested maize yields, while being less dependent on the applied fertilization (Figures 6 and 7). Utilization of N from fertilizers (R$_N$) depends on the relationship between the demand of plants for N and the amount of this nutrient originating from the applied N fertilizer [62]. A low value of R$_N$ leads to economic and ecological consequences as nitrogen that is not taken up by crops or soil microorganisms is lost due to leaching or escapes in a gaseous form [63]. Szulc et al. [64] showed that, compared to traditional fertilization, row dressing fertilization or row dressing fertilization combined with top-dressing raised the values of such parameters as the uptake and utilization of N, contribution of N fertilizer into total nitrogen uptake, as well as the AE and PE of nitrogen utilization. Halvorson et al. [65] maintain that the R$_N$ by maize is reversely proportional to the soil content of available forms of this element, and ranged between 30 and 55%. Kruczek [47], who examined the effect of a dose and foliar application of N to maize, reported R$_N$ within a range of 24 up to 59%. Furthermore, Świerczewska and Sztuder [66] showed that R$_N$ varied from 38% when maize was fertilized with single NPK fertilizers up to 51% in the variant where NPK suspension was applied before sowing and UAN was delivered in a top-dressing treatment. According to Hernandez-Ramirez et al. [67], fertilization with UAN resulted in a higher N uptake and R$_N$ compared to other fertilizers used in maize cultivation. In the presence of a relatively high level of available P in soil, additional fertilization with P did not increase R$_N$ [68]. The R$_N$ from fertilizers by maize ranged from 69 to 231%, and decreased with an increasing N dose. Supplementation of S significantly increased the R$_N$ from the applied fertilizers [44].

PNB expresses the amount of N accumulated in maize grain per 1 kg of N contained in applied fertilizers. Values of PNB within 0.7 and 1.3 kg kg$^{-1}$ implicate characteristics of sustainable agriculture, whereas values above 1.3 kg kg$^{-1}$ are considered too high and suggest that maize plants exploit the soil resources of nitrogen [69]. Finally, when PNB drops below 0.7 kg kg$^{-1}$, the applied N doses are inefficient. The weather conditions, which varied between the years, meant that the PNB values were different in the subsequent years (Figure 8a). In the dry year 2015 and in the very wet 2017, the low PNB values (0.59 and 0.69 kg kg$^{-1}$, respectively) prove the low efficiency of the applied N dose. However, in the conditions that were optimal for the development of maize, such as in 2016, the PNB value reached 0.99 kg kg$^{-1}$, which proves that the applied N dose was adequate for the harvested grain yields. Fertilization only slightly differentiated the value of this indicator

(Figure 8b). On average for the three years of the experiment, depending on the applied fertilization regimes, PNB ranged from 0.72 to 0.78 kg kg$^{-1}$.

The NUE is relatively low in conventional farming systems across the world, including developed countries. Less than 50% of the N introduced to agricultural ecosystems in the form of mineral and natural fertilizers is effectively used by crops, while the remaining quantities are dispersed in the environment, causing adverse ecological changes [70]. A very broad-scale study concerning NUE was presented by Quemada et al. [71]. Based on data collected from 195 farms in Europe, mostly specializing in cereal production, the cited authors demonstrated that the average NUE was 60% (after fertilization with a dose of 176 kg ha$^{-1}$). The utilization of N from fertilizers can be greatly improved by applying a single nitrogen treatment to the plants' rhizosphere [8]. Such fertilization considerably raised the grain yield (by 7.0%) and significantly increased the utilization of nitrogen from fertilizer, thereby reducing potential N losses. Nitrogen management depending on soil can improve the NUE by maize in connection with both agricultural practices and optimal regional nitrogen management, especially when annual weather conditions and sowing density data are also taken into consideration [7].

A large influx of N from external sources can maximize the maize yield, but it may also cause a decrease in the efficiency of how this nutrient is utilized [72]. These researchers concluded that a decrease in maize yields following a decreased N dose was mostly due to the worse efficiency of using solar radiation. Hence, it was suggested to establish the minimum efficient N dose in fertilizers that would not have a negative effect on maize grain yield. Maintaining high maize yields while reducing the negative impact on the natural environment needs further research on integrating weather forecasts with nitrogen supply in soil [73].

## 5. Conclusions

The highest grain yields were harvested after the application of UAN + S/UAN + Mg, and after the pre-sowing and top-dressing application of UAN or UAN + P (Medium). Thus, these fertilizers can be recommended for use in growing maize for grain. The best results with respect to HI index were achieved in the following fertilization variants: pre-sowing and top-dressing application of UAN + P (Medium), pre-sowing UAN and top-dressing of urea, and pre-sowing UAN + P (Starter) and top-dressing UAN + S. Maize was able to utilize nitrogen from mineral fertilizers (RN) better after the fertilization with UAN + P (Starter) pre-sowing and UAN + Mg top-dressing. HIN varied from around 55 to around 60%, and the best effect was achieved when maize was fertilized with UAN + P (Medium) pre-sowing and top-dressing. Considering the agricultural, economic and environmental effects, N fertilization needs to be designed according to the course of the weather in the year when maize is grown. The results of our research indicate that it is worthwhile to address the enrichment of UAN with such nutrients as P, K and Mg. In our opinion, it is also worthwhile to address research on the effects of micronutrient-enriched UAN on crops.

**Author Contributions:** Conceptualization, J.W. and S.S.; methodology, J.W. and S.S.; software, J.W. and S.S.; validation, S.S., J.W. and A.Ś.; formal analysis, J.W., S.S. and A.Ś.; investigation, J.W. and S.S.; resources, S.S. and J.W.; data curation, J.W., S.S. and A.Ś.; writing—original draft preparation, J.W., S.S and A.Ś.; writing—review and editing, S.S. and J.W.; visualization, J.W. and S.S.; project administration, S.S. and J.W.; funding acquisition, S.S. and J.W. All authors have read and agreed to the published version of the manuscript.

**Funding:** The results presented in this paper were obtained as part of a comprehensive study financed by the University of Warmia and Mazury in Olsztyn, Faculty of Agriculture and Forestry, Department of Agricultural and Environmental Chemistry (grant No. 30.610.003-110) and as part of a grant financed by GRUPA AZOTY ZAKŁADY AZOTOWE "PUŁAWY" SA, with its seat in Puławy, Al. Tysiąclecia Państwa Polskiego 13, 24-110 Puławy, on the topic "Determining the agricultural and economic efficiency of spring fertilization of winter oilseed rape, winter wheat and common maize with ammonium nitrate-urea solution enriched with macronutrients (P, Mg and S)" The project was financially supported by the Minister of Science and Higher Education from the program "Regional

Initiative of Excellence" for the years 2019–2022, Project No. 010/RID/2018/19, amount of funding 12.000.000 PLN.

**Institutional Review Board Statement:** Not applicable.

**Informed Consent Statement:** Not applicable.

**Data Availability Statement:** No new data were created or analyzed in this study. Data sharing is not applicable to this article.

**Conflicts of Interest:** The authors declare no conflict of interest.

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
