# Peer review of "Yield and Nitrogen Status of Maize (Zea mays L.) Fertilized with Solution of Urea—Ammonium Nitrate Enriched with P, Mg or S"

_agronomy, doi:10.3390/agronomy12092099_

Round 1

Reviewer 1 Report

The described method is quite well known and successfully used as a simple qualitative test. The entire article lacks comparisons and statistical analyzes. He proposed to the former authors that they make at least simple correlations, which would certainly enrich the prepared article. The research material was maize, however, apart from a short sentence, it was not described, for example, the technology of its production, I recommend that the authors read the article: Energies 2021, 14, 170. https://doi.org/10.3390/en14010170 

The article contains statistical data, but they are at the primary level. The statistical analysis of the factors was not supported by any post-hoc test (eg Duncan), which would allow any more advanced conclusions to be drawn. The manuscript has very mediocre results.

Author Response

Thank you very much for reviewing our manuscript.

Please be advised that the correlations have been calculated and included in the manuscript.

Regarding the comment "The statistical analysis of the factors was not supported by any post-hoc test (eg Duncan), which would allow any more advanced conclusions to be drawn" we inform you  that such analysis have been done. However, this was not explicitly stated in our manuscript, it is now. Differences between the means were compared with the Tukey’s HSD post-hoc test.

We would like to inform you that the data regarding the cultivation of corn has been supplemented are included in the manuscript (verse 105 through verse 131).

We do not agree with the opinion that: "The manuscript has very mediocre results". There are no scientific papers dealing with fertilization of maize with UAN enriched with P, Mg or S.

Reviewer 2 Report

The manuscript aims to study the ‘Fertilization of maize with UAN enriched with P, Mg or S’. I have several major concerns with the manuscript which prevent me from recommending it for publication in its current situation.

The main concern is:

Title:

(1) Should be improved. Please consider rewording your title. It is general and does not give any information.

(2) All abbreviations should be defined in first mention, please revise this issue in the whole ms!

Abstract:

(3) The abstract is very poorly constructed. For example, the aim of the work, the study design and methodology, as well as the main results that were obtained to conclude their conclusion is not presented well in the abstract. This part needs to be completely re-written. I would advise the authors to re-write the abstract part focusing primarily on the foundation of the experiments they have undertaken and the main results they have obtained.

(4) All abbreviations should be defined in first mention.

Introduction:

(5) There is a lack of information about what is new. Authors should cited the newest previous work.

(6) At the end of this section, authors should illustrate what hypothesis this investigation aimed to test.

Material and methods:

(7) The presentation of the treatments is not clear. This section should be completely revised. Please rewrite the treatments you did.

(8)  In material and methods, please cite all described methods such as in Line 96.

(9) How much plants were taken for different analysis should be mentioned clearly.

(10) I wonder that the authors have not shown any growth parameters like plant height, shoot and root fresh weight, leaves area, leaves number, etc.

Results and Discussion: are poorly written and very unclear.

(11) I would like to ask authors to separate Results from the Discussion section and put it in a separate section.

(12) In Tables 3& 4, please put standard deviations or standard errors (±SE or ±SD) values.

(13) The section is more a review type of the state of the art but does not critically discuss the findings and any potential pitfalls in the experiments.

(14) Authors should discuss how their results fill the gap of previous studies.

This section is too general without establishing the relationship among the parameters. More importantly, why and how each treatment increases or decreases a parameter ……what are the possible mechanisms, must be discussed with proper references.

Conclusion:

(15) The concluding section needs to be improved as it is too lengthy. It must be shortened and client-oriented. Authors should include specific results of their research, which extend the current state of knowledge. Add the significance and future prospect of the study.

Linguistic quality:

(16) The language quality is so poor and this paper must be edited by professional English editor.

Author Response

Thank you very much for reviewing our manuscript.

We kindly inform you about the changes made to ours in accordance with your comments.

(1) The title of the manuscript has been changed.

(2) The problem of abbreviations - corrected according to the reviewer's suggestion.

(3) The abstract has been corrected according to the comments of the reviewer.

(4) The problem of abbreviations - corrected according to the reviewer's suggestion.

(5) We would like to kindly inform you that in our manuscript we cited works from recent years (almost 30%).

(6) The hypothesis is provided at the end of this chapter.

(7) We have slightly improved the methodology and, in our opinion, it is correctly written..

(8) All methods are given according to the comments of the reviewer.

(9) Regarding the comment - "How much plants were taken for different analysis should be mentioned clearly." - we inform you that this information is already included in the manuscript.

(10) We would like to kindly inform you that the aim of the research was not the influence of UAN with an additive on the biometric features of maize, the research focused on the yield and nitrogen management of this plant.

(11 - 14) As suggested by the reviewer, the Results from the Discussion section has been separated. In tables 3 and 4 already provide values for standard errors (± SE). We would like to kindly explain that the weather conditions changed very strongly during our three-year research. This determined the maize yield and also changed all nitrogen management indicators of this plant. This has been very strongly emphasized in our manuscript. We have already indicated that there are no studies on the combined use of UAN with P, Mg or S. This is a novelty in our research. In addition, we enriched the manuscript with correlation analysis.

(15) The summary section has been revised and shortened.

(16) The quality of the language was improved by a professional English editor.

Round 2

Reviewer 2 Report

The comments have been mostly addressed except: the abbreviations should be defined in first mention, please revise this issue in the title, abstract, and the whole ms!